# SUBGRAPH ATTENTION FOR NODE CLASSIFICATION AND HIERARCHICAL GRAPH POOLING

## ABSTRACT

Graph neural networks have gained significant interest from the research community for both node classification within a graph and graph classification within a set of graphs. Attention mechanism applied on the neighborhood of a node improves the performance of graph neural networks. Typically, it helps to identify a neighbor node which plays more important role to determine the label of the node under consideration. But in real world scenarios, a particular subset of nodes together, but not the individual nodes in the subset, may be important to determine the label of a node. To address this problem, we introduce the concept of subgraph attention for graphs. To show the efficiency of this, we use subgraph attention with graph convolution for node classification. We further use subgraph attention for the entire graph classification by proposing a novel hierarchical neural graph pooling architecture. Along with attention over the subgraphs, our pooling architecture also uses attention to determine the important nodes within a level graph and attention to determine the important levels in the whole hierarchy. Competitive performance over the state-of-the-arts for both node and graph classification shows the efficiency of the algorithms proposed in this paper.

## 1 INTRODUCTION

Graphs are the most suitable way to represent different types of relational data such as social networks, protein interactions and molecular structures. Typically, A graph is represented by $G = (V, E)$, where $V$ is the set of nodes and $E$ is the set of edges. Further, each node $v_i \in V$ is also associated with an attribute (or feature) vector $x_i \in \mathbb{R}^D$. Recent advent of deep representation learning has heavily influenced the field of graphs. Graph neural networks are developed to use the underlying graph as a computational graph and aggregate node attributes from the neighbors of a node to generate the node embeddings (Kipf & Welling, 2017; Niepert et al., 2016). Different types of attribute aggregation approaches are proposed in the literature (Hamilton et al., 2017). Attention mechanisms on graphs show promising results for both node classification (Veličković et al., 2018) and graph classification (Lee et al., 2019; 2018) tasks. There are different ways to compute attention mechanisms on graph. Veličković et al. (2018) compute attention between a pair of nodes in the immediate neighborhood to capture the importance of a node on the embedding of the other node by learning an attention vector. Lee et al. (2018) compute attention between a pair of nodes in the neighborhood to guide the direction of a random walk in the graph for graph classification. Lee et al. (2019) propose self attention pooling of the nodes which is then used to capture the importance of the node to generate the label of the entire graph.

Most of the attention mechanisms developed in graph literature use attention to derive the importance of a node or a pair of node for different tasks. But in real world situation, calculating importance up to a pair of nodes is not adequate. Often due to the presence of a substructure in the vicinity of a node $v$ determines its role (or label) in the graph, or determines the label of the entire graph. But the influence of each node individually from that substructure to the node $v$ may not be significant. In Figure 1, each node (indexed from $a$ to $g$) in the small synthetic graph can be considered as an agent whose attributes determine its opinion (1:positive, 0: neutral, -1: negative) about 4 products. Suppose the graph can be labelled +1 only if there is a subset of connected (by edges) agents who jointly have positive opinion about all the product. In this case, the blue shaded connected subgraph $(a, b, c)$ is important to determine the label of the graph. Please note, attention over the pairs (Veličković et al., 2018) is not enough as $(a, b)$ cannot make the label of the graph +1 by itself. Also

multiple layers of such attention may not work as the aggregated features of a node get corrupted after the feature aggregation by the first attention layer. With this motivation, we develop a novel attention mechanism in the graph which operates in the subgraph level in the vicinity of a node. We call it subgraph attention mechanism and use it for both node classification and graph classification tasks, which we define formally next.

**Node classification**: Given a graph $G = (V, E)$ with each node $v_i$ associated with an attribute vector $x_i \in \mathbb{R}^D$, and a subset of nodes $V_s \subseteq V$ with each node $v_i \in V_s$ labelled with $y_i \in \mathcal{L}_n$ (set of discrete labels for the nodes of the graph), the task is to predict the label of a node $v_j \in V_u = V \setminus V_s$ using the structure and the node attributes of the entire graph and the node labels from $V_s$. Essentially, this leads to learning a function $f_n : V \mapsto \mathcal{L}_n$ for the given graph $G$.

**Graph Classification**: Given a set of $M$ graphs $\mathcal{G} = \{G_1, G_2, \cdots, G_M\}$, and a subset of graphs $\mathcal{G}_s \subseteq \mathcal{G}$ with each graph $G_i \in \mathcal{G}_s$ are labelled with $Y_i \in \mathcal{L}_g$ (the subscript $g$ stands for 'graphs'), the task is to predict the label of a graph $G_j \in \mathcal{G}_u = \mathcal{G} \setminus \mathcal{G}_s$ using the structure of the graphs and the node attributes, and the graph labels from $\mathcal{G}_s$. Again, this leads to learning a function $f_g : \mathcal{G} \mapsto \mathcal{L}_g$. Here, $\mathcal{L}_g$ is the set of discrete labels for the graphs.

**Contributions**: In this paper, we propose a novel attention mechanism (called subgraph attention) for graph neural networks, which is based on the importance of a subgraph of dynamic size to determine the role of a node in the graph. To validate the efficiency of the mechanism, we propose a node classification algorithm (referred as SubGatt). Further, we propose a graph classification algorithm (referred as SubGattPool) using subgraph attention layer. SubGattPool employs a mixture of hierarchical and global pooling strategies on graph. Thorough experimentation on real world graphs shows the merit of the proposed algorithms over the state-of-the-art.

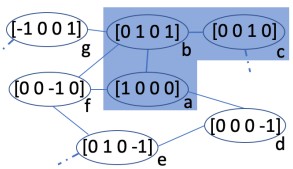

Figure 1: Example to motivate subgraph attention

## 2 RELATED WORK

A survey on network representation learning and graph neural networks can be found in Wu et al. (2019). Here we briefly discuss some more prominent approaches for node classification and graph classification. Kipf & Welling (2017) propose a version of graph convolution network (GCN) which learns a weighted mean of neighbor node attributes to find the embedding of a node by minimizing the cross entropy loss for node classification. Different extensions of GCN are available in literature for inductive learning (Hamilton et al., 2017) and link prediction (Zhang & Chen, 2018). Veličković et al. (2018) propose GAT which uses attention mechanism to learn the importance of a node to determine the label of another node in the neighborhood of it in the graph convolution framework. Recently, a GCN based unsupervised approach (DGI) is proposed (Veličković et al., 2019) by maximizing mutual information between patch representations and corresponding high-level summaries of a graph.

On the other hand, graph classification is a classical problem in machine learning. Graph kernel (Vishwanathan et al., 2010) based approaches remain to be the state-of-the-art for graph classification for long time. Yanardag & Vishwanathan (2015) propose Deep Graph Kernels which learns hidden representations of sub-structures used in graph kernels. But most of the existing graph kernels use hand-crafted features. Graph2vec (Narayanan et al., 2017) is another graph embedding technique which maximizes the likelihood of the set of subgraphs which belong to a graph. Recently, different types of graph neural networks (GNN) are proposed for graph classification. To go from node embeddings to a single representation for the whole graph, simple aggregation technique such as taking the average of node embeddings in the final layer of a GCN (Duvenaud et al., 2015) and more advanced deep learning architectures that operate over the sets (Gilmer et al., 2017; Zhang et al., 2018) have been used. Attention based graph classification technique GAM (Lee et al., 2018) is proposed, which processes only a portion of the graph by adaptively selecting a sequence of informative nodes. DIFFPOOL (Ying et al., 2018) is a recently proposed hierarchical GNN which uses a GCN based pooling to create a set of hierarchical graphs in each level. Lee et al. (2019) propose a self attention based pooling strategy which determines the importance of a node to find the label of the graph. Inspired by CapsNet (Sabour et al., 2017), Capsule GNN (CapsGNN) is proposed (Xinyi & Chen, 2019). Different extensions of GNNs, such as Ego-CNN (Tzeng & Wu, 2019) and

ChebyGIN (Knyazev et al., 2019) are proposed for graph classification. A theoretical framework to analyze the representational power of GNNs is developed by Xu et al. (2019), and a neural architecture GIN is proposed. As discussed in Section 1, attention mechanisms used in the existing GNN literature only consider self-attention or the pair-wise attention between two nodes. Recently, attention over an area for a grid structure data (for e.g., images) is proposed by Li et al. (2019) and show promising results for tasks like image captioning and machine translation. With this motivation, and to fill the research gap, we propose subgraph attention which generalizes attention to a subgraph level and show its efficiency for node and graph classification in this paper.

## 3 Subgraph Attention Mechanism

In this section, we describe the building blocks for subgraph attention layer for any arbitrary graph. A **Sub-G**raph **att**ention network (referred as SubGatt) can be built by stacking multiple layers of subgraph attention. The input to the model is an attributed graph $G = (V, E)$, where $V = \{v_1, v_2, \cdots, v_N\}$ is the set of $N$ nodes and $x_i \in \mathbb{R}^D$ is the attribute vector of the node $v_i \in V$. The output of the model is a set of node features (or embeddings) $h_i \in \mathbb{R}^K$, $\forall i \in [N]$ ($K$ is potentially different from $D$). We use $[N]$ to denote the set $\{1, 2, \cdots, N\}$ for any positive integer $N$. We define the immediate (or first order) neighborhood of a node $v_i$ as $\mathcal{N}_i = \{v_j | (v_i, v_j) \in E\}$. For the simplicity of notations, we assume an input graph $G$ to be undirected for the rest of the paper, but extending it for directed graph is straightforward.

### 3.1 Subgraph selection and Sampling

For each node in the graph, we aim to find the importance of the nearby subgraphs to that node. In general, subgraphs can be of any shape or size. Motivated by the prior works on graph kernels Shervashidze et al. (2011), we choose to consider only a set of rooted subtrees as the set of candidate subgraphs. So for a node $v_i$, any tree of the form $(v_i)$, or $(v_i, v_j)$ where $(v_i, v_j) \in E$, or $(v_i, v_j, v_k)$ where $(v_i, v_j) \in E$ and $(v_j, v_k) \in E$, and so on will form the set of candidate subgraphs of $v_i$. We restrict that maximum size (i.e., number of nodes) of a subtree is $T$. Also note that, the node $v_i$ is always a part of any candidate subgraph for the node $v_i$ according to our design. For example, all possible subgraphs of maximum size 3 for the node a in Figure 1 are: (a), (a,b), (a,d), (a,f), (a,b,c), (a,b,f), (a,b,g), (a,d,e), (a,f,e) and (a,f,b).

Depending on the maximum size ($T$) of a rooted subtree, the number of candidate subgraphs for a node can be very large. For example, the number of rooted subgraphs for the node $v_i$ is $d_{v_i} \times$
$$\sum_{v_j \in \mathcal{N}(v_i)} (d_{v_j} - 1) \times \sum_{v_k \in \mathcal{N}(v_j) \setminus \{v_i\}} |\mathcal{N}(v_k) \setminus \{v_i, v_j\}|, \text{ where } d_v \text{ is the degree of a node } v \text{ and } T = 4.$$
Clearly, computing attention over these many subgraphs for each node is computationally difficult. So we employ a subgraph subsampling technique, inspired by the node subsampling techniques for network embedding (Hamilton et al., 2017). First, we fix the number of subgraphs to sample for each node. Let the number be $L$. For each node in the input graph, if the total number of rooted subtrees of size $T$ is more than (or equal to) $L$, we randomly sample $L$ number of subtrees without replacement. If the total number of rooted subtrees of size $T$ is less than $L$, we use round robin sampling (i.e., permute all the subtrees, picking up samples from the beginning of the list; after consuming all the trees, again start from the beginning till we complete picking $L$ subtrees). For each node, sample of subtrees remains same for one epoch of the algorithm (explained in Section 3.2) and new samples are taken in each epoch. In any epoch, let us use the notation $\mathbf{S}_i = \{S_{i1}, \cdots, S_{iL}\}$ to denote the set (more precisely it is a multiset as subgraphs can repeat) of sampled subgraph for the node $v_i$.

### 3.2 Subgraph Attention Network

This subsection describes the attention mechanism on the set of rooted subtrees selected for each epoch of the algorithm. As mentioned, the node of interest is always positioned as the root of each subgraph generated for that node. Next step is to generate a feature for the subgraph. We tried different simple feature aggregations (for e.g., mean) of the nodes that belong to the subgraph as the feature of the subgraph. It turns out that concatenation of the features of nodes gives better performance. But for the attention to work, we need equal length feature vectors (the length is $TD$) for all the subgraphs. So if a subgraph has less than $T$ nodes, we append zeros at the end to assign

equal length feature vector for all the subgraphs. For example, if the maximum size of a subgraph is $T = 4$, then the feature of the subgraph $(v_i, v_j, v_k)$ is $[x_i||x_j||x_k||0] \in \mathbb{R}^{4D}$, where $||$ is the concatenation operation and $0$ is the zero vector in $\mathbb{R}^D$. Let us denote this derived feature vector of any subgraph $S_{il}$ as $\hat{x}_{il} \in \mathbb{R}^{TD}$, $\forall i \in [N]$ and $\forall l \in [L]$.

Next, we use self-attention on the features for the sampled subgraphs for each node as described here. As the first step, we use a shared linear transformation, parameterized by a trainable weight matrix $W \in \mathbb{R}^{K \times TD}$, to the feature of all the sampled subgraphs $S_{il}$, $\forall i \in [N]$ and $\forall l \in [L]$ selected in an epoch. Next we introduce a trainable self attention vector $a \in \mathbb{R}^K$ to compute the attention coefficient $\alpha_{il}$ which captures the importance of the subgraph $S_{il}$ on the node $v_i$, as follows:

$$\alpha_{il} = \frac{exp(\sigma(a^T W \hat{x}_{il}))}{\sum\limits_{l' \in [L]} exp(\sigma(a^T W \hat{x}_{il'}))} \quad , \quad h_i = \sigma\Big(\sum_{l=1}^{L} \alpha_{il} W \hat{x}_{il}\Big) \in \mathbb{R}^K \ , \quad \forall i \in [N] \qquad (1)$$

Here $\sigma()$ is a non-linear activation function. We have used Leaky ReLU as the activation function for all the experiments. $\alpha_{il}$ gives normalized attention scores over the set of sampled subgraphs for each node. We use them to compute the representation $h_i$ of a node $v_i$ as shown in Eq. 1. The main difference between our attention mechanism to that described in Veličković et al. (2018) is that we propose attention over the subgraphs, whereas they propose attention over the immediate neighboring nodes. Needless to say, one can easily extend the above subgraph attention by multi-head attention by employing few independent attention mechanisms of Eq. 1 and concatenate the resulting representations (Vaswani et al., 2017). This completes one full subgraph attention layer. We can stack such multiple layers to design a full subgraph attention network (SubGatt).

### 3.3 Learning and Node Classification

To use subgraph attention network for node classification, we use a softmax (or logistic sigmoid, depending on the number of classes to predict) as the final layer on top of the node representations. We use standard back propagation algorithm with ADAM optimization to learn the parameters of SubGatt by minimizing the cross entropy loss. For one subgraph attention layer, we just need to learn the linear transformation matrix $W \in \mathbb{R}^{K \times TD}$ and the attention vector $a \in \mathbb{R}^K$. Hence the number of parameters to learn is limited for a smaller value of $T$ (maximum size of a subgraph).

**Runtime Complexity of a subgraph attention layer**: First we need to compute the set of possible subgraphs (i.e., rooted subtrees of maximum size $T$) and their respective features for each node, once for the whole dataset. Computing set of all subgraphs and their features for a node takes $d^T$, where $d$ is the average degree of the nodes in the graph. We set $T = 4$ for our experiments and average degree of the nodes in real life graphs are small as the networks are highly sparse in nature. Next, in each epoch of SubGatt, we sample $L$ subgraphs for each node and compute Eq. 1. Hence, total runtime for each epoch of subGatt takes $O(NLKTD + Nd^T)$, which is linear with the number of nodes, for a sparse graph. Experimentally, we observe SubGatt to converge fast on all the datasets.

## 4 Hierarchical Graph Pooling with Subgraph Attention

As discussed in Section 2, GNN architectures combined with different pooling mechanisms got promising results for graph classification. The main challenge here is to obtain a single representation of the entire graph which can be used as the features for graph classification in an end-to-end fashion. Global pooling mechanisms, where the node representation are averaged or summed to obtain a graph representation for classification is proposed early in the literature (Duvenaud et al., 2015). Recently, hierarchical graph pooling becomes popular among the researchers, where a graph is converted to a graph of sub-communities in the next level, and then further to a graph of communities and till it becomes only a single node (Ying et al., 2018; Lee et al., 2019). Our approach, though inspired from the hierarchical pooling proposed by Ying et al. (2018), combines both global and hierarchical pooling by attention in different levels in the graph. We refer the proposed graph classification architecture by SubGattPool (**Sub-G**raph and sublevel **att**ention based **Pool**ing mechanism), which is described below.

As shown in Figure 2, there are $R = 4$ different levels of the graph in the architecture. The first level is the input graph and the last level is a single node whose features represent the whole input

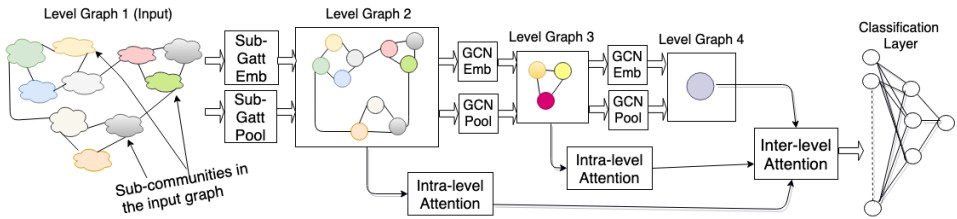

Figure 2: Architecture of SubGattPool Network for graph classification

graph. Let us denote these level graphs (i.e., graphs at different levels) by $G^1, \cdots, G^R$. There is a GNN layer between the level graph $G^r$ (i.e., the graph at level $r$) and the level graph $G^{r+1}$. This GNN layer comprises of an embedding layer which generates the embedding of the nodes of $G^r$ and a pooling layer which maps the nodes of $G^r$ to the nodes of $G^{r+1}$. We refer the GNN layer between the level graph $G^r$ and $G^{r+1}$ by $r$th layer of GNN, $\forall r = 1, 2, \cdots, R - 1$. The last level graph $G^R$ contains only one node, whose feature summarizes the entire input graph. Pleas note, number of nodes $N_1$ in the first level graph depends on the input graph, but we keep the number of nodes $N_r$ in the consequent level graphs $G^r$ ($\forall r = 2, \cdots, R$) fixed for all the input graphs (in a graph classification dataset), which help us to design a shared attention mechanism, as discussed later.

Let us assume that any level graph $G^r$ is defined by its adjacency matrix $A_r \in \mathbb{R}^{N_r \times N_r}$ and the feature matrix $X_r \in \mathbb{R}^{N_r \times K}$ (except for the level 1 graph, which is the input graph and its feature matrix $X_r \in \mathbb{R}^{N_r \times D}$). Naturally, $A_1$ and $X_1$ are given to the GNN. The $r$th embedding layer and the pooling layer are defined by:

$$Z_r = \begin{cases} \text{SubGatt}_{embed}(A_r, X_r) \, , \ r = 1 \\ \text{GCN}_{r,embed}(A_r, X_r) \, , \ r > 1 \end{cases} \qquad P_r = \begin{cases} \text{softmax}(\text{SubGatt}_{pool}(A_r, X_r)) \, , \ r = 1 \\ \text{softmax}(\text{GCN}_{r,pool}(A_r, X_r)) \, , \ 1 < r \le R - 1 \end{cases}$$
(2)

Here, $Z_r \in \mathbb{R}^{N_r \times K}$ is the embedding matrix of the nodes of $G^r$. The softmax after the pooling is applied row-wise. $(i, j)$th element of $P_r \in \mathbb{R}^{N_r \times N_{r+1}}$ gives the probability of assigning node $v_i^r$ in $G^r$ to node $v_j^{r+1}$ in $G^{r+1}$. Based on these, the graph $G^{r+1}$ is constructed as follows,

$$A_{r+1} = P_r^T A_r P_r \in \mathbb{R}^{N_{r+1} \times N_{r+1}} \ \text{ and } \ X_{r+1} = P_r^T Z_r \in \mathbb{R}^{N_{r+1} \times K}$$
(3)

The matrix $P_r$ contains information about how nodes in $G^r$ are mapped to the nodes of $G^{r+1}$, and the adjacency matrix $A_r$ contains information about the connection of nodes in $G^r$. Eq. 3 combines them to generate the connections between the nodes (i.e., the adjacency matrix $A_{r+1}$) of $G^{r+1}$. Node feature matrix $X_{r+1}$ of $G^{r+1}$ is also generated similarly. Please note that we use SubGatt only after the input graph (i.e., level graph 1) as the presence of some critical structures are high there. In contrast, other level graphs $G^r$ ($r > 1$) have probabilities as edge weights and nodes are connected to many other nodes with weights. Hence, finding subgraphs there would be computational expensive.

**Intra-level attention layer**: As observed in Lee et al. (2019), hierarchical GNNs often suffer because of the loss of information in various embedding and pooling layers, from the input graph to the one node summarizing the entire graph. To alleviate this problem, we propose to use attention mechanisms again, to combine features from different level graphs of our hierarchical architecture. We consider level graphs $G^2$ to $G^R$ for this, as their respective numbers of nodes are same across all the graphs in a dataset. We introduce *intra-level attention layer* to obtain a global feature for each level graphs $G^r$, $\forall r = 2, \cdots, R - 1$ (since, $G^r$ has only one node, so no feature aggregation is required). More precisely, we use the convolution based self attention within the level graph $G^r$ as:

$$e_r = \text{softmax}(\widetilde{D}_r^{-\frac{1}{2}} \widetilde{A}_r \widetilde{D}_r^{-\frac{1}{2}} X_r \theta) \ \in \mathbb{R}^{N_r} \ \text{ and } \ x^r = X_r^T e_r \ \in \mathbb{R}^K$$
(4)

Here, the softmax to compute $e_r$ is taken so that a component of $e_r$ becomes the normalized (i.e., probabilistic) importance of the corresponding node in $G^r$. $\widetilde{A}_r = A_r + I_{N_r}$ is the adjacency matrix with added self loops of $G^r$. $\widetilde{D}$ is the diagonal matrix of dimension $N_r \times N_r$ with $\widetilde{D}(i, i) = \sum_{j=1}^{N_r} \widetilde{A}_{ij}$.

$\theta \in \mathbb{R}^K$ is the trainable set of parameters of intra-level attention, which is shared across all the level graphs $G^r$, $\forall r = 2, \cdots, R - 1$. Intuitively, $\theta$ contains the importance of individual attributes and

| Dataset | #Nodes | #Labels | #Attributes |
|---------|--------|---------|-------------|
| Cora | 2708 | 7 | 1433 |
| Citeseer | 3312 | 6 | 3703 |
| Pubmed | 19717 | 3 | 500 |

(a) Datasets for node classification

| Dataset | #Graphs | #Max Nodes | #Labels | #Attributes |
|---------|---------|------------|---------|-------------|
| MUTAG | 188 | 28 | 2 | NA |
| PROTEINS | 1113 | 620 | 2 | 1 |
| NCI1 | 4110 | 111 | 2 | NA |
| NCI109 | 4127 | 111 | 2 | NA |
| IMDB-BINARY | 1000 | 136 | 2 | NA |
| IMDB-MULTI | 1500 | 89 | 3 | NA |

(b) Datasets for graph classification

Table 1: Different datasets used in our experiments

the components of $N_r$ dimensional $X_r\theta$ gives the same for each node. Finally, multiplying that with $\widetilde{D}_r^{-\frac{1}{2}}\widetilde{A}_r\widetilde{D}_r^{-\frac{1}{2}}$ produces the (normalized) importance of a node based on its own features and the features of immediate neighbors (for one layer of intra-level attention). Hence, $x^r$, which is a $K$ dimensional representation of the level graph $G^r$, is a sum of the features of the nodes weighted by the respective normalized node importance. Please note, the impact from the first few level graphs becomes noisy due to too many subsequent operations in a hierarchical pooling method like in Ying et al. (2018). But representing level graphs separately by the proposed intra-level attention make their impact more prominent.

**Inter-level attention layer**: After the application of intra-level attention layer, we have one vector representation $x_r \in \mathbb{R}^K$ from $G^r, \forall r = 2, \cdots, R-1$, and the feature vector $Z_R \in \mathbb{R}^K$ (from Eq. 2). Let's set $x_R = Z_R$. Here, we aim to get the final representation of the input graph from $x_2, \cdots, x_R$; in contrast to a hierarchical pooling which just considers $Z_R$ to be the final representation. We introduce *inter-level attention layer*, which takes $x_2, \cdots, x_R$ as input and generates $x^G \in \mathbb{R}^K$ as the final graph representation to be fed to a neural classifier. We again use a self-attention as follows:

$$\widetilde{e} = \text{softmax}(X_{inter}\widetilde{\theta}) \in \mathbb{R}^{R-1} \text{ and } x^G = X_{inter}^T\widetilde{e} \in \mathbb{R}^K \tag{5}$$

$X_{inter}$ is the $R-1 \times K$ dimensional matrix whose rows correspond to $x^r$ (the output of intra-level attention layer for $G^r$), $r = 2, \cdots, R$. $\widetilde{e} \in \mathbb{R}^K$ is a trainable self attention vector. Similar to Eq. 4, softmax is taken to convert $\widetilde{e}$ to a probability distribution of importance of different graph levels. Finally, the vector representation $x^G$ of the input graph is computed as a weighted sum of representations of different level graphs $G^2, \cdots, G^R$. $x^G$ is fed to a classification layer of the GNN, which is a dense layer followed by a softmax to classify the entire input graph in an end-to-end fashion. This completes the construction of SubGattPool architecture.

First layer of SubGattPool consists of an embedding SubGatt network and a pooling SubGatt network, which have a total of $O(KTD)$ trainable parameters. Consequent layers of SubGattPool have GCN as embedding and pooling layers, which have a total of $O(RKD)$ parameters. Total number of parameters for $R-2$ intra-level attention layers is $O(K)$, as $\theta \in \mathbb{R}^K$ is shared across the level graphs. Finally the inter-level attention layer has $O(K)$ parameters. Hence, total number of parameters to train in SubGattPool network is $O(KTD+RKD)$, which is independent of both the average number of nodes and the number of graphs in the dataset. We use ADAM on the cross-entropy loss of graph classification to train these parameters.

## 5 EXPERIMENTAL EVALUATION

We conduct thorough experimentation in this section. Performance on node classification shows the merit of the subgraph attention (SubGatt) network alone. Whereas, performance on graph classification shows the combined effect of different building blocks of SubGattPool. We also conduct a small experiment with graph visualization to show the merit of individual components of SubGattPool.

### 5.1 PERFORMANCE ON NODE CLASSIFICATION

**Datasets and Baseline Algorithms**: We use three popular and publicly available citation networks, where each node has a class label, for node classification. The details of the datasets are given in Table 1a (https://linqs.soe.ucsc.edu/data). We use the following state-of-the-art neural network based algorithms as baselines. node2vec (Grover & Leskovec, 2016) (skip-gram

| Dataset | node2vec | GCN | GAT | DGI | **SubGatt** |
|---------|----------|-----|-----|-----|---------|
| Cora | 80.9±1.0 | 84.3±1.8 | 81.2±2.3 | 71.0±1.8 | **87.3±0.9** |
| Citeseer | 55.2±0.7 | 71.3±1.1 | **75.6±0.8** | 68.0±1.3 | 74.3±0.5 |
| Pubmed | 80.1±0.7 | 82.8±0.5 | 79.0±0.3 | 78.8±0.7 | **87.0±0.2** |

Table 2: Classification accuracy (%) of different algorithms for node classification.

based) and DGI (GNN based) are two efficient unsupervised node embedding algorithms. We use SVM to classify the nodes on the embeddings generated by node2vec and DGI. As discussed in Section 2, GCN and GAT are semi supervised GNN based node classification algorithms.

**Experimental Setup**: We use 80% of the total node labels for training the algorithms and remaining 20% for testing. Further, 10% of the training set is used as validation. The splits are made randomly. We conduct all the experiments 10 times and reported the mean and standard deviation of classification accuracy on the test sets for all the algorithms. We set all the hyper parameters of SubGatt based on the convergence of loss in the training and the accuracy on the validation set. We use 2 layers of subgraph attention in SubGatt network for all the datasets of node classification. Learning rate is fixed to 0.001 in ADAM. Maximum size ($T$) of a subgraph is kept as 4 and the number of subgraphs sampled ($L$) for each node in an epoch of SubGatt is 32. We use dropout and L2 normalization on the parameters of SubGatt. Embedding dimension ($K$) is set to 64 for all the datasets. For baselines, we use the best parameterization on the datasets mentioned in their respective papers.

**Performance Analysis**: Table 2 shows that the proposed algorithm SubGatt is able to outperform all the baselines on Cora and Pubmed datasets in terms of average classification accuracy, whereas outperformed narrowly by GAT on Citeseer. Interestingly, the standard deviation of SubGatt is always less compared to the baselines on all the datasets which shows the robustness of the algorithm.

## 5.2 Performance on Graph Classification

**Datasets and Baseline Algorithms**: We used 4 bioinformatics graph datasets and 2 social network datasets to evaluate the performance of graph classification. Table 1b contains the high-level summary of the datasets (`https://bit.ly/2CH22ia`). We use a diverse set of baseline algorithms for graph classification. Deep graph kernel (DGK) and graph2vec are unsupervised way to generate a graph kernel and graph embedding respectively. We use SVM on top of them to predict the graph labels. We also use 5 recently proposed state-of-the-art GNN based graph classification techniques: GAM, DIFFPOOL, CapsGNN, GIN and SAGPool. We use publicly available implementation for all the baselines to generate the results.

**Experimental Setup**: We vary the training size from 15% to 60% for graph classification, and use the remaining for test. All of such splits are done randomly. We use 10% of training as the validation data. Experiments are repeated 10 times and the average classification accuracy and the standard deviation are reported. For SubGattPool, we kept the number ($R$) of level graphs (including the input graph) to be 3 for all the datasets except PROTEINS. For PROTEINS, we set $R$ to be 4 as the maximum number of nodes in PROTEINS is much higher than the other datasets. The SubGatt layer in SubGattPool samples 12 subgraphs per node for each epoch, with maximum subgraph size as 4. We set the pooling ratio (defined as $\frac{N_r}{N_{r+1}}$, $\forall r \neq 1, R-1$) to be 0.25 for PROTEINS and 0.1 for the other smaller datasets. We vary the learning rate of ADAM from 0.0001 to 0.001 based on the convergence of cost in the training. Embedding dimension (K) is fixed to 64 for all the datasets.

**Performance Analysis**: Table 3 shows that SubGattPool performs the best on MUTAG dataset for graph classification for all the training sizes. For the other datasets, the best performing algorithm vary over different training sizes. We can see that SubGattPool is always close to the best performing one. For most of the cases, the improvement of performance with increasing training is not very significant. This shows most of these algorithms saturate early with varying training sizes.

## 5.3 Incremental Effects of the Components of SubGattPool

SubGattPool network has two novel components. First, SubGatt layer, which has already performed good for node classification, and second, intra and inter-level attention layers which make SubGattPool a mixture of both global and hierarchical pooling strategy. Figure 3 shows the graph

| Algorithm | Training Size(%) | | | |
|---|---|---|---|---|
| | 15 | 30 | 45 | 60 |
| DGK | 73.4±1.3 | 75.8±1.2 | 76.2±0.9 | 76.5±1.5 |
| graph2vec | 64.6±3.6 | 67.7±3.1 | 67.1±3.1 | 68.0±5.7 |
| GAM | 67.0 ±0.6 | 67.2±2.7 | 66.4±3.1 | 64.3±5.5 |
| DIFFPOOL | 61.±15.2 | 77.8±5.5 | 78.5±1.2 | 76.9±5.6 |
| CapsGNN | 52.5±2.6 | 69.5±5.0 | 67.5±3.6 | 66.8±5.5 |
| GIN | 70.2±5.4 | 76.5±3.8 | 77.5±3.2 | 76.8±4.8 |
| SAGPool | 65.9±0.9 | 68.0±2.6 | 64.0±1.6 | 65.2 ±2.7 |
| **SubGattPool** | **80.8±2.6** | **80.9±2.6** | **81.1±1.2** | **80.3±3.1** |

(a) MUTAG

| Algorithm | Training Size(%) | | | |
|---|---|---|---|---|
| | 15 | 30 | 45 | 60 |
| DGK | 65.8±0.9 | 66.3±1.3 | 67.9±1.8 | 68.2±1.6 |
| graph2vec | 62.9±1.5 | 65.6±1.1 | 66.4±1.7 | 67.7±2.1 |
| GAM | 61.1±2.0 | 61.3 ±1.8 | 59.8±1.4 | 59.6±1.8 |
| DIFFPOOL | 67.5±2.1 | 67.0±2.2 | 68.6±1.4 | 68.1±3.1 |
| CapsGNN | 63.2±2.5 | 66.4± 2.7 | 68.9± 1.6 | 70.1±1.0 |
| GIN | 62.2±2.8 | 63.0±2.1 | 62.9±1.6 | 63.5±3.3 |
| SAGPool | 68.6±4.3 | 70.7±3.2 | 70.8 ±1.2 | 68.0 ±1.0 |
| **SubGattPool** | **71.4±1.0** | **71.2±1.3** | **72.2±1.0** | **72.4±2.6** |

(b) PROTEINS

| Algorithm | Training Size(%) | | | |
|---|---|---|---|---|
| | 15 | 30 | 45 | 60 |
| DGK | 60.5±0.7 | 61.3±1.2 | 62.3±1.0 | 62.1±0.6 |
| graph2vec | 62.6±1.0 | 63.7±1.0 | 64.9±0.3 | 64.7±2.9 |
| GAM | 51.4±1.2 | 51.9±0.8 | 52.3±1.1 | 49.7±0.5 |
| DIFFPOOL | 61.0±0.9 | 62.4±1.3 | 63.2±1.6 | 64.5±1.1 |
| CapsGNN | 55.1±2.8 | 57.6±2.5 | 62.4±1.2 | 62.8±2.0 |
| GIN | 59.8±0.9 | 64.7 ± 1.2 | 65.0±0.3 | 66.5 ±1.1 |
| SAGPool | 60.5±5.6 | 66.5±1.2 | 65.1±3.8 | 65.2±2.0 |
| **SubGattPool** | **65.8 ±0.7** | **67.6±1.2** | **69.6±1.2** | **70.4± 0.7** |

(c) NCI1

| Algorithm | Training Size(%) | | | |
|---|---|---|---|---|
| | 15 | 30 | 45 | 60 |
| DGK | 55.0±1.1 | 56.1±0.9 | 58.0±0.8 | 57.7±0.3 |
| graph2vec | 62.1±1.1 | 64.3±0.7 | 65.7±0.8 | 65.9±2.2 |
| GAM | 50.5±0.6 | 51.1±1.3 | 52.1±1.4 | 51.3±1.3 |
| DIFFPOOL | 60.1±1.3 | 62.2±1.2 | 62.7±1.3 | 63.1±1.5 |
| CapsGNN | 51.9±1.5 | 60.5±2.8 | 61.8±1.6 | 63.1±1.2 |
| GIN | 55.3±0.4 | 60.7±2.3 | 63.4±0.9 | 64.4±1.5 |
| SAGPool | 57.7±4.6 | 66.1±2.1 | 66.9±1.4 | 66.5±1.0 |
| **SubGattPool** | **64.6±0.9** | **67.2±0.6** | **67.8±0.6** | **67.7±1.0** |

(d) NCI109

| Algorithm | Training Size(%) | | | |
|---|---|---|---|---|
| | 15 | 30 | 45 | 60 |
| DGK | 60.1±2.5 | 61.0±1.8 | 62.3±2.1 | 63.5±1.9 |
| GAM | 48.9± 0.2 | 48.3±0.4 | 50.2±2.1 | 50.9±1.7 |
| DIFFPOOL | 61.0±1.4 | 64.3±1.8 | 64.3±2.3 | 65.4±2.1 |
| CapsGNN | 49.4±0.2 | 48.9±0.1 | 48.0±0.4 | 47.9±0.2 |
| GIN | **66.4±2.2** | 64.0±2.9 | **68.2±1.5** | 68.1±2.0 |
| SAGPool | 58.7 ±5.1 | 59.8 ±7.7 | 62.6 ±5.6 | 65.0±3.9 |
| **SubGattPool** | 64.1 ±1.1 | **65.4 ±1.9** | 67.5 ±1.3 | **68.2±0.6** |

(e) IMDB-BINARY

| Algorithm | Training Size(%) | | | |
|---|---|---|---|---|
| | 15 | 30 | 45 | 60 |
| DGK | 40.1±2.4 | 42.0±1.3 | 42.9±0.9 | 42.6±1.1 |
| GAM | 32.7± 0.2 | 32.4±0.6 | 32.3±0.5 | 32.0±0.6 |
| DIFFPOOL | 42.3±2.2 | 44.2±2.3 | 45.1±2.2 | 45.6±3.6 |
| CapsGNN | 35.2±1.4 | 36.2±1.5 | 37.1±1.2 | 37.7±1.3 |
| GIN | 43.7±1.8 | **46.8±2.1** | **47.8±1.2** | 47.5±2.6 |
| SAGPool | 42.4±1.3 | 43.7±2.1 | 44.8±1.5 | 45.8±1.1 |
| **SubGattPool** | **44.2±1.4** | 46.2±0.4 | 46.5±1.2 | **47.7±1.1** |

(f) IMDB-MULTI

Table 3: Classification accuracy (%) of graph classification on multiple datasets.

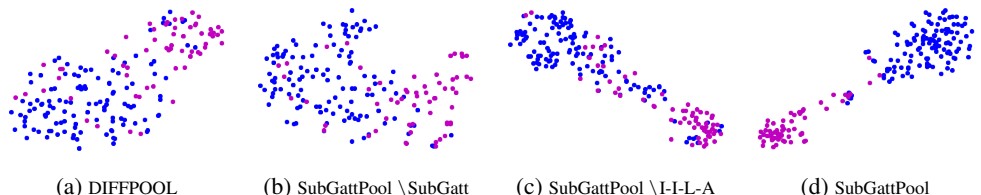

(a) DIFFPOOL     (b) SubGattPool \SubGatt     (c) SubGattPool \I-I-L-A     (d) SubGattPool

Figure 3: t-SNE visualization of the graphs from MUTAG (different colors show different labels of the graphs) by the representations generated by: (a) DIFFPOOL; (b) SubGattPool, but the SubGatt embedding and pooling layers being replaced by GCN; (c) SubGattPool without intra and inter layer attention; (d) the complete SubGattPool network. Compared to (a), there are improvement of performances for both the SubGatt layer and intra/inter-level attention individually. Finally different classes are separated most by SubGattPool which again shows the merit of the proposed algorithm.

visualization using t-SNE (van der Maaten & Hinton, 2008) on MUTAG dataset. Fig. 3b and 3c explain the incremental improvement of performance by only intra and inter level attentions, and SubGatt layer respectively. Finally, Fig. 3d shows the best performance by SubGattPool, which combines all these components into a single network.

## 6 DISCUSSION AND FUTURE WORK

In this paper, we introduce subgraph attention mechanism over graph structured data. We also propose a novel graph pooling algorithm which uses subgraph attention, and combines different hierarchical levels of graphs by separate attention layers. Current formulation of subgraph attention is based on heuristic and backed by experimental results. So in future, we would like to theoretically examine its expressiveness power for node and graph representation. Also we would like to conduct experiments in the inductive setting for node classification with SubGatt network.

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

# A APPENDIX

## A.1 SENSITIVITY ANALYSIS OF SUBGATT NETWORK

SubGatt network has three important hyper parameters. They are: (i) Maximum size of a subgraph $(T)$, (ii) Number of subgraphs sampled per node in each epoch $(L)$ and (iii) Dimension of the final node representation or embedding $(K)$ (See Eq. 1) and (iv) Number of SubGatt layers used in the network. We conduct node classification experiments on Cora. Figure 4 shows the variation of the performance of SubGatt network for node classification with respect to all these hyper-parameters. We have shown both average node classification accuracy and standard deviation over 10 repetition for each experiment. Interestingly, the variation is less with respect to all the hyper-parameters and hence it shows the robustness of SubGatt. Please note, when we are varying one hyper-parameter of SubGatt, the values of all other hyper-parameters are fixed to the values mentioned in Section 5.1.

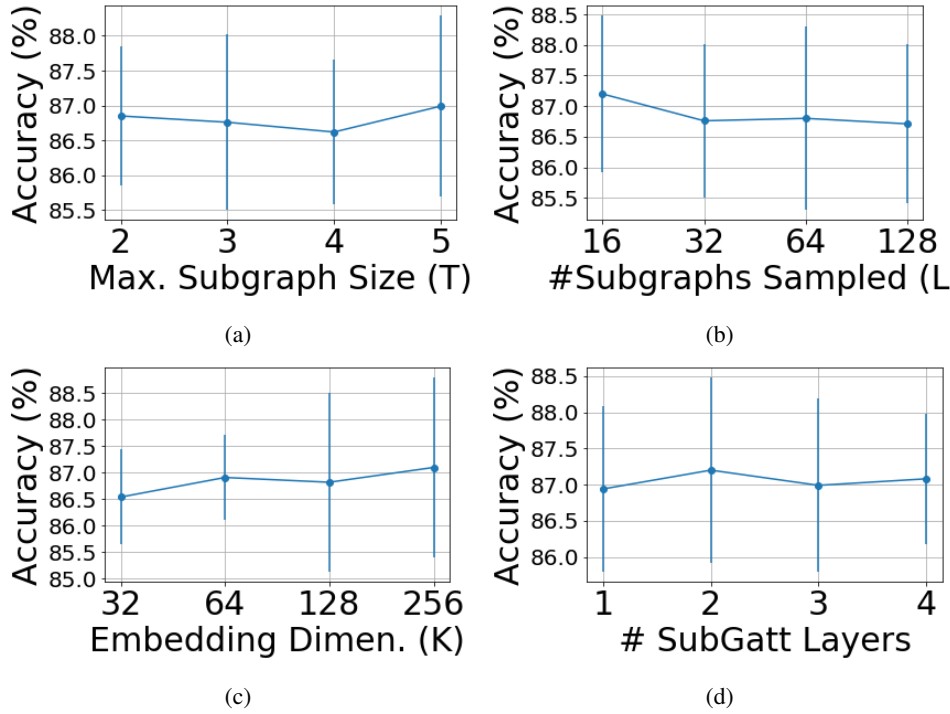

Figure 4: Sensitivity analysis of SubGatt network for node classification on Cora with respect to different hyper-parameters

## A.2 NODE CLASSIFICATION RESULTS ON THE PUBLICLY AVAILABLE SPLITS OF THE DATASETS

Some existing node classification and semi-supervised node representation techniques (Kipf & Welling, 2017; Veličković et al., 2018) use publicly available splits (into training, validation and test sets) of some popularly used datasets such as Cora, Citeseer and Pubmed to show the merit of the proposed algorithms. These splits were introduced in (Yang et al., 2016). However, these splits were created randomly, as mentioned in the cited paper: "We randomly sample 20 instances for each class as labeled data, 1000 instances as test data". Kipf & Welling (2017) introduces "an additional validation set of 500 labeled examples for hyperparameter optimization". We use exactly the same splits of the nodes as in (Kipf & Welling, 2017) and report the results in Table 4. It turns out that SubGatt is outperformed by GAT on this publicly available splits of the datasets, whereas the performance is competitive to GCN. This is slightly contradicting to what we observe in Table 2, where the performance of SubGatt is mostly superior to GCN and GAT on the random splits of the dataset into training, validation and test sets. One possible reason can be the larger training size (80% training, in which 10% was used only for validation) of the node classification experiments

| Dataset | #Train/Validation/Test | GCN | GAT | **SubGatt** |
|---------|------------------------|-----|-----|-------------|
| Cora | 140/500/1000 | 81.5 | **83.0±0.7** | 81.4 |
| Citeseer | 120/500/1000 | 70.3 | **72.5±0.7** | 70.5 |
| Pubmed | 60/500/1000 | **79.0** | **79.0±0.3** | 78.2 |

Table 4: Classification accuracy (%) of different algorithms for node classification on the publicly available node splits into training, validation and test sets.

conducted in Section 5.1. Compared to that, the publicly available split has much lesser training size and SubGatt being a more complex model than GCN and GAT, needs more training data to effectively train the parameters.

