# OpenReview forum: "Subgraph Attention for Node Classification and Hierarchical Graph Pooling"
_ICLR.cc/2020/Conference — Reject_

### Official Review · AnonReviewer2 · 2019-10-20
**Official Blind Review #2**

**Rating:** 6

**Review:**

The paper proposes a novel attention approach to graph neural networks which is applicable to both of node and graph classification. The proposed method gives an attention to a subgraph instead of a node by which importance can be controlled as a set of nodes. Further, the authors also introduce two types of attention in the hierarchical structure of the network called intra- and inter- level attention.

Considering subgraph attention would be novel and a reasonable idea. The sampling-based approach is a bit naive though it would be easy to implement. Reliability of some results are not clear for me because of the small training set.

The intra- and inter- attention approach would be a reasonable, but the relation with subgraph attention is not mentioned in my understanding. These two are independent approaches? Nothing is related to each other?

The MUTAG dataset has only 188 graphs, and so, in the smallest case in Table 3, the training data only contains 188*0.15 = 28.2 graphs. For me, learning an attention neural network with less than 30 sample is difficult to evaluate. Is there any rationale that the proposed method works on such a small dataset?

In sensitivity analysis in A.1, the performance on different max subgraph size (T) is shown, and the change of the performance is moderate. One of the main claims of the paper is that considering a subgraph (not a node) increases the performance. This results does not show the increase of the performance with the increase of the subgraph size T. Showing the performance with T = 1 can be informative to verify improvement brought by the subgraph attention.

Is Figure 3 training set or test set?

**Experience Assessment:**

I do not know much about this area.

**Review Assessment: Checking Correctness Of Derivations And Theory:**

I did not assess the derivations or theory.

**Review Assessment: Checking Correctness Of Experiments:**

I assessed the sensibility of the experiments.

**Review Assessment: Thoroughness In Paper Reading:**

I read the paper at least twice and used my best judgement in assessing the paper.

---

> ### Author Response · Authors · 2019-11-09
> **Response to Review #2**
>
> We thank the reviewer for the insightful comments. The concept of subgraph attention is proposed in a generic sense, and can be applied for both node and graph representation. Whereas, intra and inter level attention mechanisms are specific to hierarchical graph classification purpose. So subgraph attention is independent of intra and inter level attention mechanisms. However, inter-level attention is an attention mechanism over the output of intra-level attention of different levels in the hierarchical architecture of SubGattPool. All these 3 types of attention mechanisms are proposed in this work.
>
> 15% training size in MUTAG dataset leads to only 28 graphs in the training set. We are still able to produce reasonably good performance because of the two reasons. First, the number of parameters to learn in SubGattPool is small compared to many GNN based graph classification algorithms, as discussed in the last paragraph of Section 4. Hence training the SubGattPool network with a small number of graphs is possible. Second, each graph has multiple nodes and edges, and SubGattPool also exploits the structure of each graph in its learning.
>
> We thank the reviewer for bringing the point on the experiment with max subgraph size (T). When T=1, there is only one possible subgraph (i.e., rooted subtree of size 1) ‘a’ for any node ‘a’. Thus, it does not make sense to consider this case.
>
> Please note, T denotes the maximum size of a subgraph selected in SubGatt. So, a candidate subgraph can have length from 1 to T in that case. Thus, it is possible that the set of subgraphs with high attention values are the same for two different values of T (for e.g., when T=3 and when T=4). Hence, with the increasing subgraph size T, the performance may not always improve.
>
> However, to see the improvement of considering the attention mechanism over subgraphs, rather than only the immediate neighboring nodes, please compare the performance of GAT and SubGatt in Table 2 for node classification.
>
> Figure 3 shows the complete set of graphs in MUTAG dataset. We used 80% of the graphs to train the algorithms and then show the final representations (for e.g., the output of inter-level attention for SubGattPool) of all the graphs using t-SNE.

---

### Official Review · AnonReviewer1 · 2019-10-25
**Official Blind Review #1**

**Rating:** 3

**Review:**

This paper introduces a subgraph attention method for graphs. Recently, many papers have shown that attention is a very important concept. However, there was no attention method for graph input structures, while a particular subset of nodes is very crucial to make the output.

This paper first proposes the graph attention mechanism and hierarchical graph pooling idea. The attention basically subsamples subtrees so that each node can have the same number of attention candidates. Then, we can the attention network as many other papers. Experimental results show that the proposed attention based algorithm outperforms other algorithms.

I think this paper attacks a very important issue "graph attention" and have a very nice algorithm and results. Overall, my recommendation is "accept".

Cons.
It would better if the authors test some other different attention networks along with the current way.

================================================
I've read all discussions and changed my score. The novely of this work is not enough as R4 pointed out.

**Experience Assessment:**

I do not know much about this area.

**Review Assessment: Checking Correctness Of Derivations And Theory:**

N/A

**Review Assessment: Checking Correctness Of Experiments:**

N/A

**Review Assessment: Thoroughness In Paper Reading:**

N/A

---

> ### Author Response · Authors · 2019-11-09
> **Response to Review #1**
>
> We thank the reviewer for the encouraging and insightful comments. Yes, there is no existing method in the literature to determine the importance of a subset of nodes (a subgraph) to a node within the vicinity for graph representation learning, and we address this important problem in this paper.
>
> We agree with the reviewer that one can test different types of attention networks (such as the dot-product attention in Vaswani, Ashish, et al. "Attention is all you need." Advances in neural information processing systems. 2017) to find the importance of subgraphs in our framework. However, the focus of the paper was not to test multiple attention mechanisms, but to show the importance of attention over the subgraphs for node and graph classification. Also, testing with different types of attention networks needs significant changes in the architecture and implementation in the graph domain. So we would like to conduct them in a separate body of work in the future.

---

### Official Review · AnonReviewer4 · 2019-11-01
**Official Blind Review #4**

**Rating:** 1

**Review:**

This work proposes a subgraph attention mechanism on graphs. Compared to the previous graph attention layer, the node in the graph attends to its subgraph. The subgraph is represented by an aggregated feature representation with a sampled fixed-size subgraph. The methods are evaluated on both node classification and graph classification problems.

I have major concerns about the novelty, and experiments in this work.

1. The motivation is not clear. Using a subgraph or neighborhood to represent a node is reasonable. However, this work samples a subset of nodes from the one-hop neighborhood and aggregates them for attention mechanism. It is very similar to a GCN + GAT. The sampling process even loses some neighborhood information in the graph.

2. The experimental setups are very strange. In Table 2, the methods are compared to GCN and GAT on node classification problems. The performance of GAT is too low and even lower than that reported in GAT. Can authors explain this? It is highly recommended to use the same experimental settings as in GCN and GAT. The same problem exists in Table 3. Can authors provide a performance comparison based on the same settings in GIN?

3. The performance improvements are very unstable and marginal. In Table 3, the proposed methods can not compete with previous methods especially on large datasets like IMDB-MULTI. I wonder how the proposed methods perform on very large datasets such as reddit.

4. Can authors provide comparisons with a simple GCN+GAT?

**Experience Assessment:**

I have published one or two papers in this area.

**Review Assessment: Checking Correctness Of Derivations And Theory:**

I carefully checked the derivations and theory.

**Review Assessment: Checking Correctness Of Experiments:**

I carefully checked the experiments.

**Review Assessment: Thoroughness In Paper Reading:**

I read the paper thoroughly.

---

> ### Author Response · Authors · 2019-11-09
> **Response to Review #4**
>
> We thank the reviewer for the insightful comments. Please note that the subgraphs selected can have different sizes. The only restriction is that the maximum size of any subgraph (or the length of a rooted subtree) is a hyper-parameter and fixed. The novelty of the work lies in the design of a subgraph attention mechanism which is proposed for the first time in the graph representation literature. Also, our graph classification architecture, which consists of the subgraph attention, intra-level attention and inter-level attention is another novel aspect of the paper. We also conduct a short ablation study in Section 5.3 to show the effect of these components for graph representation.
>
> To address Point 1 of R#4, we do NOT sample a subset of nodes from one hop neighborhood of a node. Rather, we sample a subset of subgraphs (in the form of rooted subtrees as discussed in Section 3.1) for each node. The self-attention is also computed over the set of subgraphs. In GAT, the self attention is calculated on the set of neighbor nodes. Please note, computing attention scores over the immediate neighbors and use neighborhood feature aggregation in multiple layers can capture weighted information beyond the immediate neighborhood, but the propagated features would be corrupted due to information aggregation in multiple stages. We also explain this in the second paragraph of Section 1 with the help from Figure 1. For example, the attributes of node ‘c’ cannot be propagated as it is to node ‘a’ in a GAT/GCN type of framework, because of the intermediate attribute aggregation (averaging) happened at node ‘b’ in Figure 1.
>
>
> To address Point 2 of R#4, our node classification experimental setting is different from GCN or GAT papers. As explained in Section 5.1 of the paper, we use 80% of the total node labels for training the algorithms and remaining 20% for testing. Further, 10% of the training set is used as validation. The splits are made randomly. We conduct all the experiments 10 times and reported the mean and standard deviation of classification accuracy on the test sets for all the algorithms. Similarly, experiments on graph classifications are also conducted over multiple splits. This set up also evaluates the robustness of different algorithms for node and graph classification. Same splits are used for the baselines and our proposed algorithms to ensure fairness.
>
> However, as suggested by the reviewer, we will update the results on the set up used in GCN and GAT for node classification within the author rebuttal period. For graph classification, mean accuracy and standard deviation based on 10-fold cross validation are reported in the GIN paper.
>
>
> To address Point 3 of R#4, we agree that SubGattPool is not always the best performing graph classification under all the training sets. But, overall its performance is competitive and often better on most of the datasets. Standard deviation on the performance of SubGattPool is also less or comparable to other baselines, which shows the stability of the algorithm. Also note that the selected baseline algorithms, such as DIFFPOOL, GIN and SAGPool are all recently proposed and state-of-the-art GNN based graph classification algorithms.
>
>
> To address Point 4 of R#4, we are not sure about the architecture the reviewer meant by GCN+GAT. GAT uses self attention over the immediate neighborhood and then does neighborhood feature aggregation. The second part is similar to GCN. So we are not sure if the reviewer indicates GAT by GCN+GAT. We have compared the performance of SubGatt with GAT and GCN in Table 2 of the paper.

---

> > ### Author Response · Authors · 2019-11-15
> > **Graph classification experiments on the setup used in GIN**
> >
> > We perform 10 fold cross validation (9 folds for training and 1 fold for validation) as done in GIN paper. Table below summarizes average accuracies on 10 folds and the respective standard deviation on MUTAG, NCI1, PROTEINS, IMDB-BINARY, IMDB-MULTI datasets, for both in GIN (we have always taken the best performer among GIN-0 and GIN-\epsilon, directly from their paper) and our proposed algorithm SubGattPool.
> >
> >
> > DATASETS               GIN         SubGattPool
> >
> > MUTAG                 90.0±8.8      90.3±4.2
> > PROTEINS            76.2±2.8      78.6±3.9
> > NCI1                     82.7±1.6      83.1±1.5
> > IMDB-BINARY    75.1±5.1      73.0±5.4
> > IMDB-MULTI      52.3±2.8      50.3±3.3
> >
> > Compared to Table 3 in our paper, the average accuracy numbers increase for both GIN and SubGattPool as the experiments are conducted on 10-fold cross validation here. Similar to Table 3 of the main paper, SubGattPool outperforms GIN on MUTAG, PROTEINS and NCI1; and is outperformed on IMDB-BINARY and IMDB-MULTI. We did not show the results on NCI109 as the GIN results were not available for this dataset. We can also observe that relative performance of GIN is better in this set up. This is because GIN is theoretically as powerful as WL test of isomorphism in terms of representational power, and that is the maximum a GNN can achieve. However, Table 3 of our main paper shows the generalization of the performance of the GNNs for graph classification on the test set (we used a separate validation set consisting of 10% of the total training set). There the relative performance of SubGattPool is better than the rest.

---

### Author Response · Authors · 2019-11-15
**Summary of the Changes in the Revised Draft**

We want to thank all the reviewers for their insightful comments to improve the quality of our paper. We have addressed individual reviewers in the respective comment sections. Here we summarize the major changes made in the revised (latest) version of our paper.

1. We have added new baseline algorithms GAM and CapsGNN for graph classification in Table 3. We have also improved the graph classification performance of SubGattPool by adding L2 and dropout regularizers on the pooling layers.  In Table 3, SubGattPool performs the best on the first four datasets and for the last two datasets, GIN and SubGattPool perform better than the other baselines. So overall, the performance of SubGattPool turns out to be the best or highly competitive over all the state-of-the-art graph classification algorithms. We have also updated the published code of SubGattPool to ensure the reproducibility of the results.

2. There was some confusion in the information provided in the description of the datasets for graph classification in Table 1b. We have updated them appropriately.

3. We have added a new section A.2 in the Appendix to discuss the node classification performance of SubGatt and comparison on the publicly available node splits of the datasets, as asked by Reviewer #4.

As asked by Reviewer #4, we have also added the graph classification results in the set up used in GIN, in the respective comment section.

---

### Decision · Program_Chairs · 2019-12-19

**Decision:**

Reject

**Comment:**

Initially, two reviewers gave high scores to this paper while they both admitted that they know little about this field. The other review raised significant concerns on novelty while claiming high confidence. During discussions, one of the high-scoring reviewers lowered his/her score. Thus a reject is recommended.